# Activation of PERK Contributes to Apoptosis and G_2_/M Arrest by Microtubule Disruptors in Human Colorectal Carcinoma Cells

**DOI:** 10.3390/cancers12010097

**Published:** 2019-12-30

**Authors:** Ming-Shun Wu, Chih-Chiang Chien, Ganbolor Jargalsaikhan, Noor Andryan Ilsan, Yen-Chou Chen

**Affiliations:** 1Division of Gastroenterology, Department of Internal Medicine, Wan Fang Hospital, Taipei Medical University, Taipei 11031, Taiwan; mswu@tmu.edu.tw; 2Division of Gastroenterology and Hepatology, Department of Internal Medicine, School of Medicine, College of Medicine, Taipei Medical University, Taipei 11031, Taiwan; 3Integrative Therapy Center for Gastroenterologic Cancers, Wan Fang Hospital, Taipei Medical University, Taipei 11031, Taiwan; 4Department of Nephrology, Chi-Mei Medical Center, Tainan City 710, Taiwan; ccchien58@yahoo.com.tw; 5Department of Food Nutrition, Chung Hwa University of Medical Technology, Tainan 71703, Taiwan; 6International MS/PhD Program in Medicine, College of Medicine, Taipei Medical University, Taipei 11031, Taiwan; ganbolor.j@gmail.com (G.J.); noorandryanilsan@gmail.com (N.A.I.); 7Liver Center, Ulaanbaatar 14230, Mongolia; 8Department of Medical Laboratory Technology, STIKes Mitra Keluarga, Bekasi 17113, West Java, Indonesia; 9Graduate Institute of Medical Sciences, College of Medicine, Taipei Medical University, Taipei 11031, Taiwan; 10Cancer Research Center and Orthopedics Research Center, Taipei Medical University Hospital, Taipei 11031, Taiwan; 11Cell Physiology and Molecular Image Research Center, Wan Fang Hospital, Taipei Medical University, Taipei 11031, Taiwan

**Keywords:** protein kinase RNA-like endoplasmic reticular kinase, microtubule disrupters, taxol, nocodazole, human colon carcinoma cells

## Abstract

Microtubule-targeting agents (MTAs) are widely used in cancer chemotherapy, but the therapeutic responses significantly vary among different tumor types. Protein kinase RNA-like endoplasmic reticular (ER) kinase (PERK) is an ER stress kinase, and the role of PERK in the anticancer effects of MTAs is still undefined. In the present study, taxol (TAX) and nocodazole (NOC) significantly induced apoptosis with increased expression of phosphorylated PERK (pPERK; Tyr980) in four human colon cancer cell lines, including HCT-15, COLO205, HT-20, and LOVO cells. Induction of G_2_/M arrest by TAX and NOC with increases in phosphorylated Cdc25C and cyclin B1 protein were observed in human colon cancer cells. Application of the c-Jun N-terminal kinase (JNK) inhibitors SP600125 (SP) and JNK inhibitor V (JNKI) significantly reduced TAX- and NOC-induced apoptosis and G_2_/M arrest of human colon cancer cells. Interestingly, TAX- and NOC-induced pPERK (Tyr980) protein expression was inhibited by adding the JNK inhibitors, SP and JNKI, and application of the PERK inhibitor GSK2606414 (GSK) significantly reduced apoptosis and G_2_/M arrest by TAX and NOC, with decreased pPERK (Tyr980) and pJNK, phosphorylated Cdc25C, and Cyc B1 protein expressions in human colon cancer cells. Decreased viability by TAX and NOC was inhibited by knockdown of PERK using PERK siRNA in COLO205 and HCT-15 cells. Disruption of the mitochondrial membrane potential and an increase in B-cell lymphoma-2 (Bcl-2) protein phosphorylation (pBcl-2; Ser70) by TAX and NOC were prevented by adding the PERK inhibitor GSK and JNK inhibitor SP and JNKI. A cross-activation of JNK and PERK by TAX and NOC leading to anti-CRC actions including apoptosis and G_2_/M arrest was first demonstrated herein.

## 1. Introduction

Colorectal cancer (CRC) is the third most common diagnosis and second deadliest malignancy for both sexes combined [1]. Many therapeutic strategies, such as surgery, chemotherapy, and radiotherapy, are used to treat CRC, and radiotherapy combined with chemotherapy is the standard of care in locally advanced rectal cancer in the setting of neoadjuvant treatment, but the outcome remains suboptimal for advanced cases. Immune checkpoint inhibitors (ICIs) appear promising for chemotherapy-refractory CRC, and the modern aspect of CRC treatment is immunotherapy with PD-1 inhibitors, nivolumab, and pembrolizumab, which currently constitute the new standard of care as treatment of chemotherapy-refractory microsatellite-instability-high (MSI-high)/mismatch repair-deficient (MMR-d) CRC. However, only a minority of CRCs have presented with MSI-high and MMR-d [2,3]. Mitosis is a multiphase mechanism, and mitotic spindles made up of microtubules are the main effector of mitosis. Aberrations of microtubule organization generate aneuploid daughter cells that are genomically unstable, leading to the apoptosis of tumor cells [4]. Microtubule-targeting agents (MTAs) were developed to induce stabilization or destabilization of microtubules, leading to blockage of the deregulated mitotic process and induction of apoptosis of cancer cells, and have been extensively used in clinical cancer treatment [5,6]. Although MTAs have been applied as cancer therapeutics for several decades, drug resistance is one of the key factors obstructing the clinical applicability of MTAs. Therefore, the antitumor mechanisms of MTAs deserve to be further investigated.

Based on their action mechanisms, MTAs are classified into two groups: one group stabilizes microtubules and includes taxol (TAX), and the other group destabilizes microtubules and includes colchicine and nocodazole (NOC) [7]. Both TAX and NOC have been extensively used in cancer studies, and previous reports have demonstrated the antitumor actions of TAX and NOC in various cancers [8,9]. NOC is a microtubule-disrupting agent shown to arrest the cell cycle at the G_2_/M phase and trigger apoptosis in various cells [10,11,12]. TAX is a chemotherapeutic drug due to its antineoplastic effects in a variety of tumors. Unlike NOC, TAX-stabilized microtubules lead to blockage of the cell cycle at the G_2_/M phase and also at the G_0_/G_1_ transition phase [13,14]. TAX was also reported to induce apoptosis in human HT29 CRC cells as well as other cells [15,16,17,18,19].

A great deal of effort has been devoted to elucidating signaling pathways that mediate the biological activities of TAX and NOC. TAX induces apoptosis in various cells mediated by different signal transduction pathways, including the phosphoinositide 3-kinase (PI3K)/AKT pathway, the epidermal growth factor (EGF) receptor pathway, and the mitogen-activated protein kinase (MAPK) pathway. Han et al. [20] indicated that NOC induced apoptosis of Jurkat cells via activation of p38 MAPK. Guo et al. [21] reported that NOC weakly activated p38 MAPK but inhibited tumor necrosis factor (TNF)-α-induced p38 MAPK activation in Rat-1 cells [21]. Protein kinase RNA (PKR)-like endoplasmic reticular (ER) kinase (PERK), a type of ER stress sensor, is a serine/threonine kinase that regulates the survival and death of cells [22]. Activation of PERK leading to induction of eukaryotic translation initiation factor 2α (eIF2α) phosphorylation was considered to be cytoprotective [23]; however, several previous studies indicated that phosphorylation of eIF2α was able to increase activating transcription factor 4 (ATF4) protein expression to initiate proapoptotic signaling [24,25]. Our previous studies demonstrated that the natural chemical evodiamine (EVO) effectively induced apoptosis and G_2_/M arrest with increased PERK protein phosphorylation in various cancer cells, including human ovarian cancer cells, glioblastoma cells, and renal cell carcinoma cells [26,27]. It would be valuable to understand the role of PERK in MTA-induced anticancer actions. In the present study, TAX and NOC reduced the viability of four human CRC cells via apoptosis and G_2_/M arrest, and involvement of PERK and c-Jun N-terminal kinase (JNK) activation leading to increased phosphorylation of the B-cell lymphoma-2 (Bcl-2) protein and disruption of the mitochondrial membrane potential (MMP) is demonstrated herein.

## 2. Materials and Methods

### 2.1. Cells

Four human CRC cell lines, including COLO205, HT-29, HCT-15, and LOVO, were purchased from the American Type Culture Collection (Manassas, VA, USA). Cells cultured in RPMI containing 10% heat-inactivated fetal bovine serum (FBS; Gibco/BRL, Grand Island, NY, USA) supplemented with antibiotics (100 U/mL penicillin A and 100 U/mL streptomycin) were maintained in a 37 °C humidified incubator containing 5% CO_2_. Cells were used between passages 18 and 30 for all experiments. After confluence, cells were seeded onto 6 cm dishes for further experiments.

### 2.2. Materials

TAX, NOC, nitroblue tetrazolium (NBT), SP600125 (SP), and 5-bromo-4-chloro-3-indolyl phosphate (BCIP) were purchased from Sigma (St. Louis, MO, USA). All chemicals were dissolved in dimethyl sulfoxide (DMSO), and the final concentration of DMSO in each treatment was <0.5%. Antibodies of PERK, JNK, cyclin B1 (cycB1), Cdc25C, Bcl-2, and α-tubulin (α-TUB) and the JNK inhibitor V (JNKI) were obtained from Santa Cruz Biotechnology (Santa Cruz, CA, USA). Antibodies of phosphorylated (p)JNK, pPERK, and pBcl-2 were obtained from Cell Signaling Technology (Beverly, MA, USA). Small interfering (si)RNA and control siRNA were obtained from Santa Cruz Biotechnology.

### 2.3. Western Blotting

Cells lysates were prepared by suspending cells in RIPA buffer, and equal amounts of protein were prepared, separated on sodium dodecylsulfate (SDS)-polyacrylamide mini gels, and transferred to Immobilon polyvinylidene difluoride membranes (Millipore, Bedford, MA, USA). Membranes were incubated at 4 °C with 1% bovine serum albumin (BSA) for a further 50 min at room temperature and then incubated with the indicated antibodies overnight at 4 °C. This was followed by incubation with an alkaline phosphatase-conjugated immunoglobulin G (IgG) antibody for 1 h. Proteins were visualized by incubation with the colorimetric substrates NBT and BCIP.

### 2.4. 3-(4,5,-Dimethylthiazol)-2-yl-2,5-Diphenyltetrazolium Bromide (MTT) Assay

Cells were plated at a density of 5 × 10^4^ cells/well on 6-well plates. At the end of treatment, the supernatant was removed, and 30 μL of the tetrazolium compound, MTT, and 270 mL of fresh minimum essential medium (MEM) were added. After incubation for 2 h at 37 °C, 200 μL of 0.1 N HCl in 2-propanol was placed in each well to dissolve the tetrazolium crystals. Finally, the absorbance at a wavelength of 600 nm was recorded using an enzyme-linked immunosorbent assay (ELISA) plate reader.

### 2.5. DNA Fragmentation Assay

Cells under different treatments were collected and then lysed in 100 µL of lysis buffer (50 mM Tris at pH 8.0, 10 mM ethylenediaminetetraacetic acid (EDTA), 0.5% sodium sarcosinate, and 1 mg/mL proteinase K) for 3 h at 56 °C. Then, 0.5 mg/mL RNase A was added to each reaction for another hour at 56 °C. DNA was extracted with phenol/chloroform/isoamyl alcohol (25/24/1) before loading. Then, DNA samples were mixed with 6 µl of loading buffer (50 mM Tris, 10 mM EDTA, 1% (w/w), and 0.025% (w/w) bromophenol blue) and loaded onto 2% agarose gel containing 0.1 mg/mL ethidium bromide. The agarose gels were run at 100 V for 45 min in TBE buffer, then observed and photographed under UV light.

### 2.6. Measurement of the MMP

After different treatments, cells were incubated with 40 nM DiOC6(3) for 15 min at 37 °C, then washed with ice-cold phosphate-buffered saline (PBS) and collected by centrifugation at 500× *g* for 10 min. Collected cells were resuspended in 500 mL of PBS containing 40 nM DiOC6(3). Fluorescence intensities of DiOC6(3) were analyzed on a flow cytometer (FACScan, Becton Dickinson, San Jose, CA, USA) with respective excitation and emission settings of 484 and 500 nm.

### 2.7. Detection of G_2_/M Arrest and Hypodiploid Cells by TAX and NOC

Cells were plated on 24-well plates in duplicate, then incubated for 24 h. Media were removed, and different compounds were added to each well. Cells were treated for 12 h, and the supernatant and cells were harvested by exposing cells to a 0.25% trypsin-EDTA solution for 10 min, after which they were centrifuged, washed in PBS, and fixed in 3 mL of ice-cold 100% ethanol. All samples were incubated for 30 min at room temperature in the dark. The cell cycle distribution and hypodiploid cells were determined using a FACSan flow cytometer (FACScan, Becton Dickinson, San Jose, CA, USA) [28].

### 2.8. Statistical Analysis

Values are expressed as the mean ± standard deviation (SD) of triplicate experiments. The significance of the difference from the respective controls for each experiment was assayed using a one-way analysis of variance (ANOVA) with post hoc Bonferroni analysis when applicable, and *p*-values < 0.01 were considered statistically significant.

## 3. Results

### 3.1. Induction of Apoptosis by TAX and NOC with Increased Phosphorylation of the PERK Protein in Human CRC Cells

We first examined the effect of two MTAs (TAX and NOC) on the viability of four human CRC cell lines, including COLO205, HCT-15, LOVO, and HT-29 cells. As shown in Figure 1A, TAX and NOC treatment induced loss of DNA integrity with the occurrence of DNA ladders in these human CRC cells according to DNA agarose electrophoresis. Western blot data indicated that increased cleavage of apoptotic protein caspase-3 (Casp-3) by TAX and NOC was detected in cells. Data of the MTT assay showed that TAX and NOC were able to significantly reduce the viability of these CRC cells (Figure 1B). Increases in the levels of phosphorylated PERK protein (Thr980) were detected in TAX- and NOC-treated human CRC cells (Figure 1C). These data showed that TAX and NOC were able to induce apoptosis of human colorectal carcinoma cells with increased PERK protein phosphorylation at Thr980.

### 3.2. Induction of G_2_/M Arrest by TAX and NOC with Increased Phosphorylated Cdc25C and cycB1 in Human CRC Cells

We further examined if TAX and NOC affected cell cycle progression and altered expressions of Cdc25C and cycB1 proteins in human CRC cells. As illustrated in Figure 2A, data of a cell cycle progression analysis by flow cytometry via propidium iodide (PI) staining indicated that a significant increase in the G_2_/M ratio was detected in TAX- and NOC-treated human COLO205, HCT-15, LOVO, and HT-29 CRC cells. In the presence of TAX treatment for different times, expressions of the phosphorylated Cdc25C and cycB1 proteins increased with time in COLO205, LOVO, and HT-29 cells (Figure 2B). These effects were mediated by both TAX and NOC, although not inducing cdc2, as shown in Figure 2C.

### 3.3. Activation of JNK Is Involved in TAX- and NOC-Induced Apoptosis of Human CRC Cells

Activation of JNK was implicated in apoptosis of several cancer cell lines under chemical stimulation. Pharmacological studies using the JNK inhibitors, including SP and JNKI, were used to study the role of JNK in TAX- and NOC-induced apoptosis of human CRC cells. As shown in Figure 3A, data of the MTT assay indicated that SP (10 μM) addition significantly prevented human CRC cells from TAX- or NOC-induced cytotoxicity. Cleavage of Casp-3 proteins induced by TAX or NOC was inhibited by the addition of SP to HCT-15, LOVO, and HT-29 cells (Figure 3B). In order to identify the role of JNK, another JNK inhibitor—JNKI—was used as described in our previous study [27]. Data of the MTT assay showed that incubation of indicated CRC cells with JNKI (10 μM) significantly reduced the cytotoxicity elicited by both TAX and NOC (Figure 3C). Analysis of DNA integrity by agarose electrophoresis showed that the intensity of DNA ladders induced by TAX or NOC was blocked by the addition of SP or JNKI to COLO205 and HCT-15 cells (Figure 3D).

### 3.4. Activation of JNK in TAX- or NOC-Induced G_2_/M Arrest of Human CRC Cells

As described in Figure 4, SP and JNKI were used to examine the role of JNK activation in TAX- and NOC-induced G_2_/M arrest of human CRC cells. Western blot data indicated that the addition of SP inhibited levels of phosphorylated Cdc25C and cycB1 proteins stimulated by TAX and NOC in human CRC cells (Figure 4A). The increased percentage of cells in the G_2_/M phase by TAX and NOC was suppressed by SP addition in COLO205, HT-29, and LOVO cells (Figure 4B). As the same part of the experiment, addition of cells with JNKI exhibited a similar effect as SP on TAX- and NOC-induced G_2_/M arrest. As shown in Figure 4C,D, inhibition of TAX- or NOC-induced G_2_/M arrest and phosphorylation of Cdc25C and cycB1 proteins by JNKI was demonstrated in human CRC cells (Figure 4C,D).

### 3.5. Suppression of JNK Activation by SP or JNKI Inhibited TAX- and NOC-Induced Phosphorylated PERK Protein Expression in Human CRC Cells

The role of JNK activation in PERK protein phosphorylation by TAX and NOC was examined using SP and JNKI. As shown in Figure 5A, incubation of CRC cells, including COLO205, HCT-15, LOVO, and HT-29 cells, with the JNK inhibitor SP inhibited both JNK and PERK protein phosphorylation by TAX and NOC. In order to verify the action of JNK, JNKI was applied in this study. As shown in Figure 5B, addition of JNKI to CRC cells reduced expressions of phosphorylated JNK and PERK proteins by TAX and NOC. These results suggest that JNK activation might contribute to PERK protein phosphorylation in human CRC cells under TAX or NOC stimulation.

### 3.6. GSK2606414 (GSK), a PERK Inhibitor, Inhibited TAX- and NOC-Induced Cell Death and G_2_/M Arrest with Suppression of PERK and JNK Protein Phosphorylation in Human CRC Cells

The PERK inhibitor GSK was used to examine the role of PERK activation in TAX- or NOC-induced apoptosis and G_2_/M arrest of human CRC cells. Data of the MTT assay showed that the addition of GSK significantly protected human COLO205 and LOVO CRC cells from TAX- and NOC-induced cell death (Figure 6A). Increases in phosphorylated PERK, JNK, Cdc25C, and cycB1 proteins by TAX and NOC were inhibited by the addition of GSK to COLO205 and LOVO cells (Figure 6B). Analysis of the G_2_/M ratio by flow cytometry indicated that the increase in the G_2_/M and decrease in G1 percentage by TAX and NOC was reversed by the addition of GSK in COLO205 and LOVO cells (Figure 6C; data not shown). Knockdown of PERK by transfection of COLO205 and HCT-15 cells with PERK siRNA significantly inhibited TAX- and NOC-induced cell death and G_2_/M arrest (Figure 6D; data not shown). These results suggest that PERK activation is involved in TAX- and NOC-induced cell death and G_2_/M arrest of human CRC cells.

### 3.7. The PERK Inhibitor GSK and JNK Inhibitors SP and JNKI Decreased TAX- or NOC-Disrupted MMP with Decreased Expression of Phosphorylated Bcl-2 Protein at Ser70

We further examined the role of PERK and JNK activation in TAX- or NOC-induced disruption of MMP in human HCT-15 colon cancer cells using the PERK inhibitor GSK and JNK inhibitors SP and JNKI. Data of a flow cytometric analysis using the fluorescent mitochondrial probe DiOC6 indicated that a reduced MMP by TAX or NOC was observed in HCT-15 cells, and it was significantly blocked by the addition of the PERK inhibitor GSK and JNK inhibitors SP and JNKI (Figure 7A). Examination of total Bcl-2 and phosphorylated Bcl-2 (pBcl-2; Ser-70) expressions in CRC cells under TAX or NOC stimulation by Western blotting using specific antibodies showed that TAX and NOC did not alter levels of the total Bcl-2 protein but increased pBCl-2 (Ser-70) protein expression in these cell lines, which was suppressed by adding the PERK inhibitor GSK or JNK inhibitors SP and JNKI (Figure 7B,C).

## 4. Discussion

We explored the contribution of PERK to MTA-induced apoptosis and G_2_/M arrest of human CRC cells. In the presence of TAX or NOC stimulation, human CRC cells showed apoptotic characteristics, including DNA ladders, hypodiploid cells, activation of caspase-3, and disruption of the MMP in accordance with arrest of the cell cycle at the G_2_/M phase. All of the above events were abolished by blocking JNK PERK in human CRC cells. In particular, application of the PERK inhibitor GSK or PERK siRNA significantly reduced TAX- or NOC-induced anti-CRC effects such as apoptosis and G_2_/M arrest, accompanied by decreased pBcl-2 (Thr 70), pCdc25C, cycB1, and pJNK protein expressions in cells. The critical role of PERK activation in the anti-CRC effect of MTAs such as TAX and NOC via inducing apoptosis and G_2_/M arrest was demonstrated.

Both apoptosis and cell cycle arrest by TAX and NOC were found in various cells. TAX binds to tubulin and stabilizes microtubule filaments to induce mitotic arrest during the G_2_/M phase and promote apoptosis [30]. Choi and Yoo [31] indicated that TAX-induced G_2_/M arrest and apoptosis were associated with p21WAF1/CIP1 in human breast cancer cells. Roth et al. [32] indicated that TAX-induced apoptosis of human malignant glioma cells was associated with Bcl-2 protein phosphorylation but not p53 or G_2_/M cell cycle arrest. It was noted that Cdc2/cycB1 complexes and Cdc25 are known as M-phase-promoting factors (MPFs), and activation of Cdc2/cycB1 and Cdc25 trigger the switch to initiate mitosis [33,34]. Three isoforms of Cdc25 phosphatases have been identified in mammals, and Cdc25C plays an important role in controlling the G_2_/M phase. Previous studies indicated that overexpression of Cdc25C promotes phosphatase activity during the interphase and is activated at the G_2_/M transition after hyperphosphorylation [35,36]. Cho et al. [37] indicated that NOC-induced hyperphosphorylation of the Cdc25C protein contributed to M-phase arrest in HEK-293 cells. In the present study, expression levels of phosphorylated Cdc25C and cycB1 proteins increased after TAX and NOC stimulation with the occurrence of G_2_/M arrest in human CRC cells, and those events were inhibited by adding the PERK inhibitor GSK. This suggests that increased phosphorylation of Cdc25C and cycB1 by PERK activation participates in TAX- and NOC-induced anti-CRC effects.

Both proapoptotic and antiapoptotic Bcl-2 family proteins were shown to regulate intrinsic and extrinsic apoptosis. The Bcl-2 protein is one of the antiapoptotic members of the Bcl-2 family, and downregulation of the Bcl-2 protein and/or increased phosphorylation levels of the Bcl-2 protein leading to disruption of the MMP are certainly important modes of apoptosis induced by chemicals. In the presence of MTA treatment, induction of phosphorylation of the Bcl-2 protein was previously reported. Ruvolo et al. [38] found that Bcl-2 phosphorylation induced by TAX might inhibit the antiapoptotic function of Bcl-2. Brichese et al. [39] indicated that JNK activation was associated with phosphorylation of the Bcl-2 protein by TAX in HeLa cells. In human CRC cells, increased phosphorylation of the Bcl-2 protein by TAX and NOC was observed with decreased MMPs according to a flow cytometric analysis using DiOC6 as a mitochondrion probe, which was suppressed by adding the PERK inhibitor GSK. These results suggest that PERK activation is involved in TAX- and NOC-induced mitochondrial disruption and phosphorylated Bcl-2 protein expression in human CRC cells.

MAPK-mediated pathways were shown to control processes associated with antitumor actions, including apoptosis and cell cycle arrest, by different therapeutic agents in various cancer cell lines [40,41]. Activation of JNK, one of the MAPK members, contributes to apoptosis and tumor progression in conjunction with a variety of stimuli, indicating the contradictory functions of JNK [42,43,44]. Several papers suggested a critical role of JNK activation in malignancy and the poor prognosis of cancer patients [45,46]. The differential roles of JNK activation in MTA-induced anticancer effects were previously reported. Huang et al. [47] reported that colchicine induced apoptosis of HT-29 cells via JNK activation. Wang et al. [48] indicated that JNK activation was involved in TAX-induced apoptosis. Sánchez-Pérez et al. [49] showed that mitotic arrest by TAX and NOC activates JNK, leading to facilitation of TNF-related apoptosis-inducing ligand (TRAIL)-induced activation of an apoptotic pathway in human breast cancer cells [49]. However, the JNK inhibitor SP600125 did not suppress NOC-induced apoptotic events in Jurkat cells [18]. Consistent with previous results, increased JNK protein phosphorylation by TAX and NOC was detected in human CRC cells, and application of the JNK inhibitors SP600125 or JNKI protected CRC cells from TAX- and NOC-induced apoptosis and G_2_/M arrest, accompanied by decreased phosphorylation of Cdc25C and cycB1 protein expressions. These results support the important role of JNK activation in the anti-CRC actions of TAX and NOC in human CRC cells. Additionally, Cellurale et al. [50] reported that JNK is required for Ras-induced transformation of MEF. In cancer cells harboring K-Ras mutations, glutamine (Gln) deprivation induced cytotoxicity to TAX via arrested cell cycle at the S or G_2_/M phase [51]. This indicated that K-Ras-driven cancer cells overrode a late G1 checkpoint by Gln deprivation and was arrested at the S phase, exploiting the potential of metabolic changes in cancer cells for therapeutic intervention [52]. In the present study, we found a cross-activation between JNK and PERK leading to apoptosis and G_2_/M arrest by TAX and NOC in human CRC cells. Contribution of PERK/JNK to K-Ras-mediated cancer metabolism and cell cycle progression is suggested for further study.

ER stress is triggered by a disruption in ER function called the unfolded protein response (UPR), and there are three signaling pathways of eukaryotic cells initiated by ER stress sensors including PERK, IRE1, and ATF6 [53]. PERK activation may phosphorylate eIF2α and activate the downstream transcriptional factor CHOP, leading to initiation of the mitochondrial pathway of apoptosis, including activation of caspases-9 and -3. Our previous studies demonstrated that PERK activation plays important roles in chemical-induced apoptosis and cell cycle arrest in different cancer cell lines [24,26,27]. Although TAX- and NOC-induced ER stress was reported, the roles of PERK in TAX- and NOC-induced apoptosis and G_2_/M arrest are still unclear [54,55]. In the present study, both TAX and NOC induced apoptotic events and G_2_/M arrest with increased PERK protein phosphorylation (Thr 980), which were blocked by the PERK inhibitor GSK. Increased PERK protein phosphorylation was abolished by the JNK inhibitors SP600125 or JNKI in CRC cells. Critical cross-activation between JNK and PERK that contributes to apoptosis and G_2_/M arrest of human CRC cells by TAX and NOC was elucidated.

Several aspects such as CUE domain containing 2 (CUEDC2) [56] and intestinal microbiota [57] in CRC pathogenesis have been reported. Our study provides a new aspect, revealing the PERK pathway as a possible target for CRC treatment. However, the study was performed on a limited number of human CRC cells and lacked in vivo investigation. Studying the actions of PERK in different CRC cells and examining the expression of PERK in human CRC samples and CRC animal models related to their sensitivity to chemotherapy are important topics for further research.

## 5. Conclusions

In conclusion, our results demonstrated that activation of PERK and JNK contributes to apoptosis and G_2_/M arrest by TAX and NOC in human CRC cells. A tentative mechanism related to MTAs such as EVO-, TAX-, and NOC-induced anti-CRC effects, including apoptosis and G_2_/M arrest, is described in Figure 8. It indicates that TAX and NOC are able to activate JNK and PERK, which in turn promote the apoptosis and G_2_/M arrest machinery, including mitochondrial apoptotic cascades and phosphorylation of Cdc25C leading to anti-CRC actions. PERK and JNK could act as important targets for resistance to CRC therapy by targeting microtubules and deserves further exploration.

## Figures and Tables

**Figure 1 cancers-12-00097-f001:**
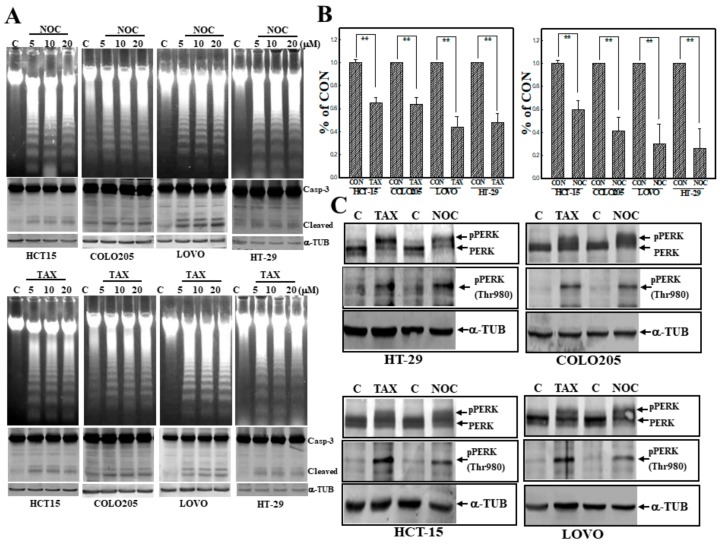
Taxol (TAX) and nocodazole (NOC) induced DNA ladders with increased protein kinase RNA-like endoplasmic reticular kinase (PERK) protein phosphorylation in human COLO205, HCT-15, LOVO, and HT-29 colorectal carcinoma (CRC) cells. (**A**) Increased DNA ladders by agarose electrophoresis and caspase-3 (Casp-3) protein cleavage by Western blotting in TAX- and NOC-treated CRC cells. Four CRC cells were treated with different concentrations (5, 10, and 20 μM) of TAX or NOC for 24 h, and DNA integrity and Casp-3 protein of cells was detected by agarose electrophoresis and Western blotting, respectively. (**B**) TAX and NOC reduced the viability of CRC cells according to an MTT assay. CRC cells were treated with TAX or NOC (10 μM) for 24 h, and the viability of CRC cells was examined by an MTT assay. Data are presented as a percentage of the control (% of CON) from three independent experiments. (**C**) TAX and NOC induced PERK protein phosphorylation at Thr980 in human CRC cells. As described in (**B**), total proteins from CRC cells under various treatments were applied for Western blotting, and levels of indicated proteins, including PERK, phosphorylated (p)PERK (Thr 980), and α-tubulin (α-TUB), were detected using specific antibodies. Each data point was calculated from three triplicate groups, and data are displayed as the mean ± SD. ** *p* < 0.01 denotes a significant difference between indicated groups.

**Figure 2 cancers-12-00097-f002:**
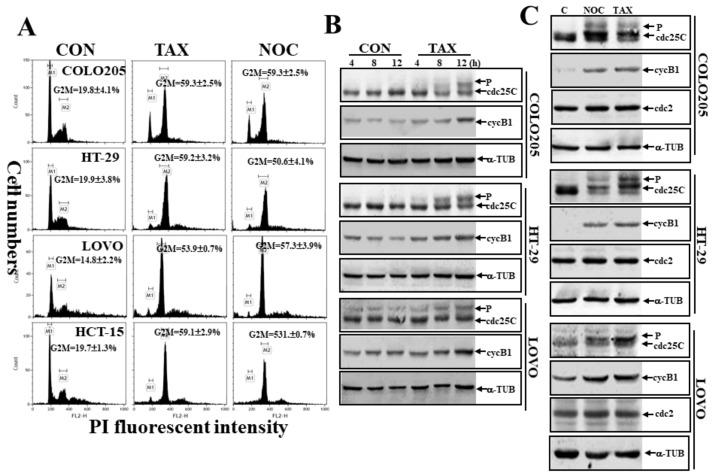
TAX and NOC induced cell cycle arrest at the G_2_/M phase with increased Cdc25C protein phosphorylation and cyclin B1 (cycB1) protein in human CRC cells. (**A**) Induction of the G_2_/M phase by TAX and NOC in human CRC cells. Cells were treated with TAX or NOC (10 μM) for 24 h, and cell cycle progression of human CRC cells under different treatments was examined by a flow cytometric analysis using propidium iodide (PI) staining. (**B**) TAX time-dependently induced phosphorylated Cdc25C and cyclin B1 protein expressions in COLO 205, HT-29, and LOVO cells. Cells were treated with TAX (10 μM) for different times (4, 8, and 12 h), and expressions of the indicated proteins were examined by Western blotting. (**C**) TAX and NOC induced phosphorylation of Cdc25C and cyclin B1, but not cdc2, in COLO 205, HT-29, and LOVO cells. Cells were treated with TAX or NOC (10 μM) for 12 h, followed by Western blotting to detect expressions of indicated proteins. Data from three independent experiments were obtained and are displayed as the mean ± SD.

**Figure 3 cancers-12-00097-f003:**
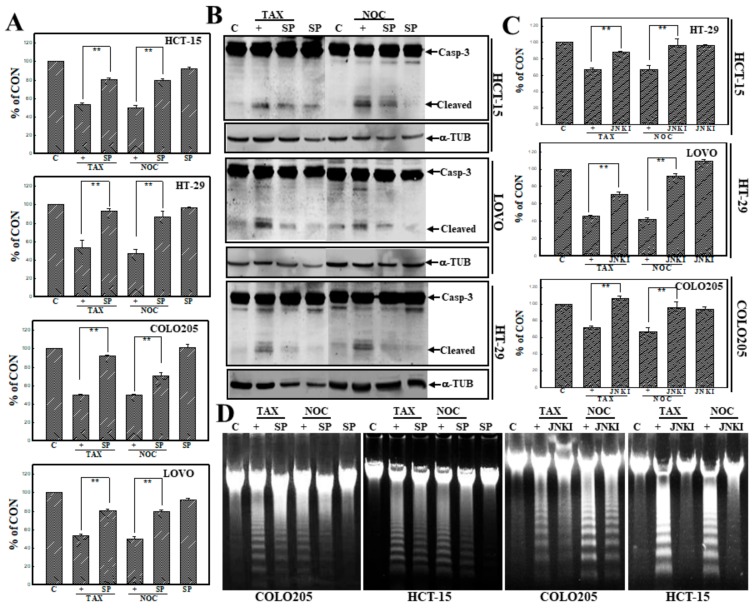
c-Jun N-terminal kinase (JNK) activation participated in TAX- and NOC-induced apoptosis of human CRC cells. (**A**) The JNK inhibitor SP600125 (SP) protected cells from TAX- and NOC-induced cytotoxicity in human CRC cells. Human CRC cells were treated with SP (10 μM) for 30 min followed by TAX or NOC (10 µM) treatment for 24 h. The viability of cells under different treatments was examined by an MTT assay, and results are expressed as a percentage of the control (% of CON). (**B**) SP inhibition of cleavage of Casp-3 protein in human CRC cells. As described in (**A**), expressions of the Casp-3 and α-TUB proteins were examined by Western blotting using specific antibodies. (**C**) Addition of JNK inhibitor V (JNKI) reversed the cytotoxicity of TAX and NOC in HCT-15, HT-29, and COLO205 cells. Cells were treated with JNKI (10 μM) for 30 min followed by TAX and NOC (10 μM) treatment for 24 h. The viability of the cells was examined by an MTT assay. (**D**) SP and JNKI inhibited TAX- and NOC-induced DNA ladders in human CRC cells. As described above, DNA integrity was examined by agarose electrophoresis. Data from three independent experiments were obtained and are displayed as the mean ± SD. ** *p* < 0.01 denotes a significant difference between indicated groups.

**Figure 4 cancers-12-00097-f004:**
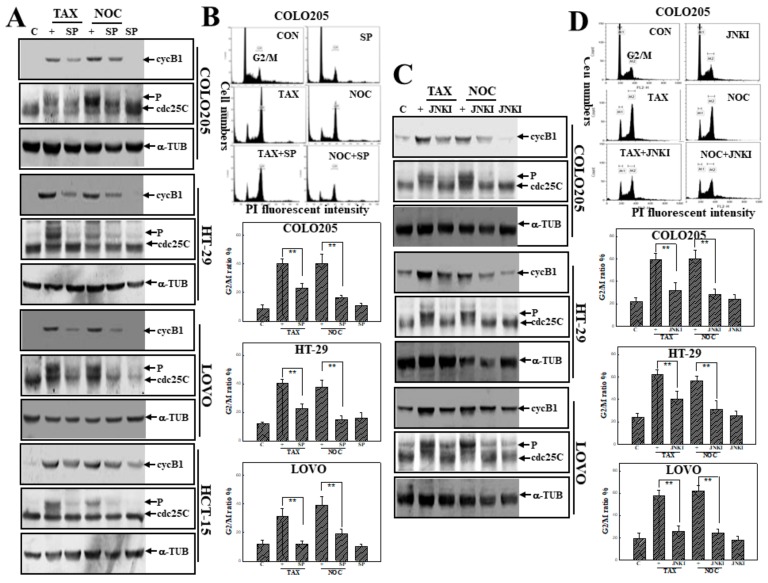
JNK activation is involved in G_2_/M arrest by TAX and NOC in human CRC cells. (**A**) The JNK inhibitor SP inhibited the increase in cyclin B1 and phosphorylated Cdc25C protein expressions by TAX and NOC in human CRC cells. As described in Figure 3A, expressions of cyclin B1, Cdc25C, and α-TUB proteins were examined by Western blotting using specific antibodies. (**B**) SP decreased TAX- and NOC-induced G_2_/M arrest in human CRC cells. As described above, the G_2_/M ratio of indicated cells by TAX and NOC with or without SP (10 μM) treatment was detected by a flow cytometric analysis using PI staining. (**C**) JNKI inhibited TAX- and NOC-induced phosphorylated Cdc25C and cyclin B1 protein expressions in the indicated human CRC cells. (**D**) As described in (**B**), JNKI reduced the G_2_/M ratio induced by TAX and NOC. Each data point was calculated from three triplicate groups, and data are displayed as the mean ± SD. ** *p* < 0.01 denotes a significant difference between indicated groups.

**Figure 5 cancers-12-00097-f005:**
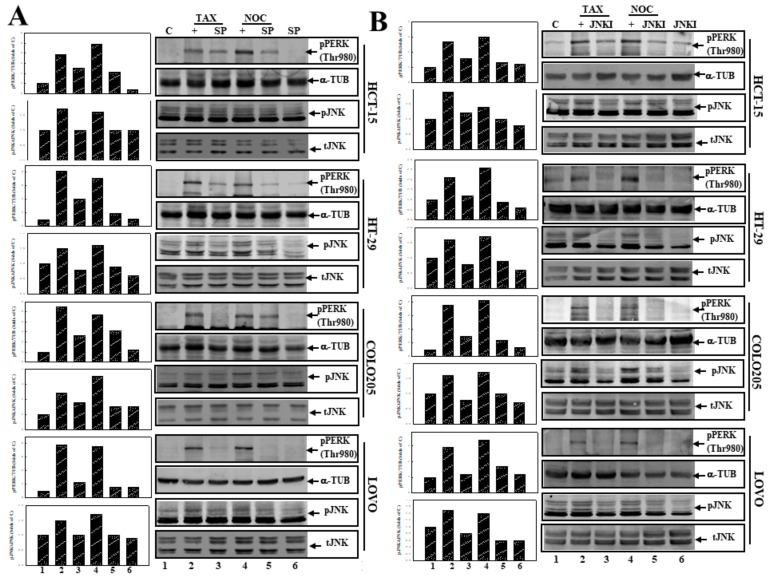
Inhibition of JNK by SP (**A**) or JNKI (**B**) inhibited PERK and JNK protein phosphorylation by TAX and NOC in human CRC cells. Cells were treated SP or JNKI (10 µM) for 30 min followed by TAX or NOC (10 μM) stimulation for 12 h. Expressions of pPERK (Thr980), pJNK, total JNK, and α-TUB proteins were examined by Western blotting using specific antibodies. (Right panel) Western blotting. (Left panel) The relative levels of pPERK (Thr980) and pJNK were normalized to α-TUB and tJNK, respectively, and quantified using Image J software [29].

**Figure 6 cancers-12-00097-f006:**
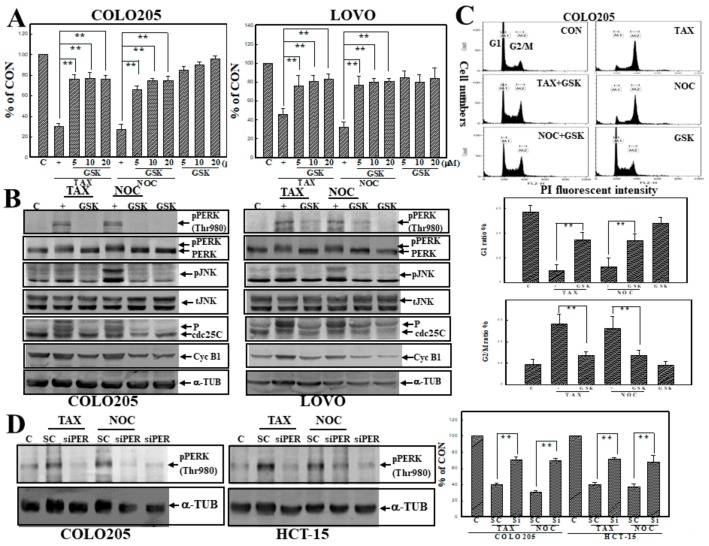
The PERK inhibitor GSK2606414 (GSK) inhibited cell death, PERK, and JNK protein phosphorylation in human CRC cells by TAX and NOC. (**A**) Both COLO205 and LOVO cells were treated with different concentrations (5, 10, and 20 μM) of GSK for 30 min, followed by TAX and NOC (10 μM) treatment for 24 h. The viability of cells under different treatments was examined by an MTT assay, and results are expressed as a percentage of the control (% of CON). (**B**) GSK inhibition of TAX- and NOC-induced cyclin B1, phosphorylated PERK, JNK, and Cdc25C protein expressions in COLO205 and LOVO cells. Both cell lines were treated with GSK (10 µM) for 30 min, followed by TAX and NOC (10 μM) stimulation for 12 h for detecting phosphorylated PERK and JNK proteins, cyclin B1, and phosphorylated Cdc25C proteins by Western blotting. (**C**) GSK reversed the increase in G_2_/M and the decrease in the G_1_ ratio by TAX and NOC in COLO205 cells. (**D**) Knockdown of PERK protein by transfection of COLO205 and HT-29 cells with PERK siRNA (Si; siPERK) inhibited TAX and NOC-induced cell death. Both cells were transfected with siPERK or scramble siRNA (SC) (1 ng), and expression of phosphorylated PERK protein and viability of both cells were examined by Western blotting (left panel) and MTT assay (right panel), respectively. Each data point was calculated from three triplicate groups, and data are displayed as the mean ± SD. ** *p* < 0.01 denotes a significant difference between the indicated groups.

**Figure 7 cancers-12-00097-f007:**
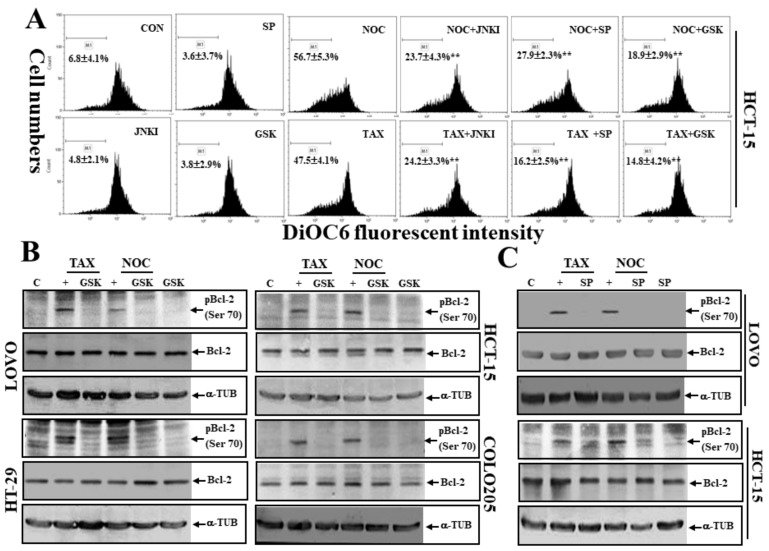
The PERK inhibitor GSK and the JNK inhibitors SP and JNKI inhibited TAX- and NOC-induced loss of the mitochondrial membrane potential (MMP) and phosphorylated B-cell lymphoma-2 (Bcl-2) (Ser-70) in human CRC cells. (**A**) HCT-15 cells were treated with GSK, SP, or JNKI (10 μM) for 30 min, followed by TAX and NOC (10 μM) stimulation for 12 h. The MMP was examined by a flow cytometric analysis via DiOC6 staining. (**B**) GSK inhibition of TAX- and NOC-induced phosphorylation of the Bcl-2 protein (pBcl-2) at Ser-70 in COLO205, HCT-15, HT-29, and LOVO cells. Cells were treated with GSK (10 μM) for 30 min, followed by TAX or NOC stimulation for 12 h. (**C**) The JNK inhibitor SP decreased TAX- and NOC-induced phosphorylation of the Bcl-2 protein (pBcl-2) at Ser-70 in HCT-15 and LOVO cells. Both cell lines were treated with SP (10 μM) for 30 min, followed by TAX or NOC stimulation for 12 h. Levels of Bcl-2, pBcl-2, and α-TUB proteins were examined by Western blotting using specific antibodies. Each data point was calculated from three triplicate groups, and data are displayed as the mean ± SD. ** *p* < 0.01 denotes a significant difference from NOC- and TAX-treated groups, respectively.

**Figure 8 cancers-12-00097-f008:**
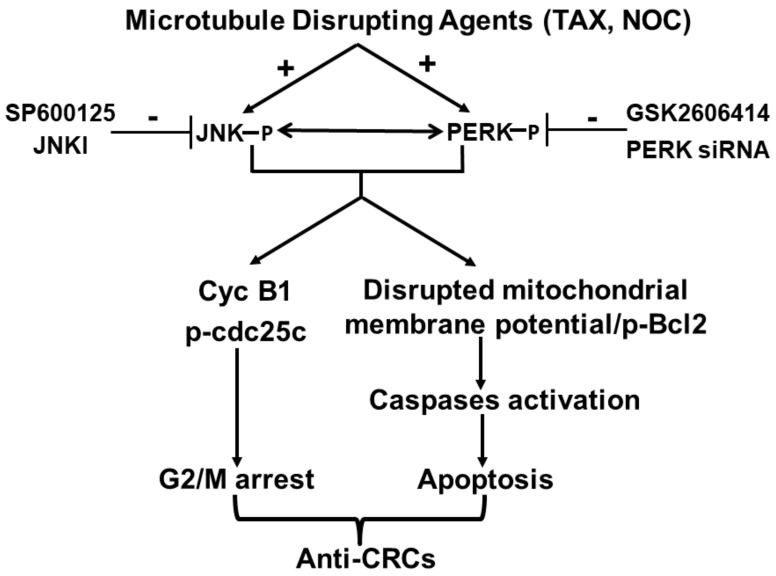
A tentative mechanism for anti-CRC actions of the microtubule-disrupting agents TAX and NOC.

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
