# Peer review of "Activation of PERK Contributes to Apoptosis and G2/M Arrest by Microtubule Disruptors in Human Colorectal Carcinoma Cells"

_cancers, 2019, doi:10.3390/cancers12010097_

Round 1
Reviewer 1 Report
I’ve enjoyed reading this well written, thorough, important and relevant original article. By focusing on the antitumoric role of microtubule-targeting agents (MTAs) in colorectal cancer (CRC) cells, the authors reveal an interesting insight for possible therapeutic targets of CRC.
The title of the paper precisely states the final conclusions in a comprehensive manner. The abstract is intelligible accurately describing the objectives and the results obtained. The introduction provides a generalized background of the topic that gives the reader an appreciation of the role of MTAs in cancer. The methods that were used in the study are valid and appropriate for the experiment the researchers carried out. They can also be duplicated because the process of each method was stated in the paper with clarity. The experimental design they used was appropriate to the objective of their study because they were able to have a productive flow of the subsequent methods involved, yielding solid results. Moreover, the several figures greatly aid the visualization of the findings in a more understandable format. The discussion greatly summarizes the results and associates them with appropriate references regarding each specific gene. Furthermore, the literature cited is adequate and relevant to the study.
I have few minor comments that may further improve the current manuscript before acceptance of publication:
Line 60: "Colorectal cancer (CRC) is the second leading diagnosed cancer". Actually, CRC is the third most common diagnosis and second deadliest malignancy for both sexes combined [1].
Line 61: "surgery and chemotherapy are used to treat CRC". Radiotherapy combined with chemotherapy is the standard of care in locally advanced rectal cancer in the setting of neoadjuvant treatment, while another modern aspect of CRC treatment is immunotherapy with PD-1 inhibitors, nivolumab and pembrolizumab, which currently constitute the new standard of care as treatment of chemotherapy-refractory MSI-high/MMR-d CRC [2].
Line 89: "Phosphorylated" should be changed to "phosphorylation".
Line 164: Here the TAX and NOC are referred as microtubule-interfering agents (MIAs), whereas in the Introduction they are referred as microtubule-targeting agents (MTAs). While both terms are true and correct, one of the two should be applied in the whole manuscript for the sake of cohesion.
Line 181-183: "In the same ... CRC cells". It should be mentioned that these effects are mediated by both TAX and NOC, although not inducing cdc2, as shown in Figure 2C.
Line 200: Replace "An" with "The".
Line 223: "was reduced". More accurate to state that it "was reversed", since the reduction is profound only in G2/M ratio.
Line 239: “suppressed by adding the PERK inhibitor GSK, or JNK inhibitor SP and JNKI (Fig. 7B, C)”. Although it is described here, the effect of JNKI is not depicted in Figure 7.
Line 293: It is not JNK siRNA, but JNKI that reversed TAX- or NOC-induced apoptosis in CRC cells.
Line 451-452 and 464: “CRCs” should be replaced by “CRC cells”.
Line 466 and 469: HT-29, not HCL-15 is depicted in Figure 2B,C.
Line 491: “increased in” should be replaced by “the increase in”.
Line 508: HCT-115 should be replaced by HCT-15 in Figure 6
Line 518: “cyc25C” should be replaced by “cdc25C”.
Line 532-534: The legend of Figure 7B lacks of description of the corresponding results.
Furthermore, in the Discussion section two small paragraphs should be added; one describing the limitations of the study, and one further expanding suggestions for future research.
All in all, this study gives a new aspect in research of CRC, revealing PERK pathway as possible target in CRC treatment.
References
Xiao JB, Leng AM, Zhang YQ, Wen Z, He J, Ye GN. CUEDC2: multifunctional roles in carcinogenesis. Front Biosci (Landmark Ed). 2019 Mar 01;24:935-946. Koliarakis, I.; Psaroulaki, A.; Nikolouzakis, T.K.; Kokkinakis, M.; Sgantzos, M.N.; Goulielmos, G.; Androutsopoulos, V.P.; Tsatsakis, A.; Tsiaoussis, J. Intestinal microbiota and colorectal cancer: A new aspect of research. JBUON. 2018, 23, 1216–1234.Author Response
Response to Reviewer 1
Thank you for reviewer’s comments. Responses have been described below. All changes in the revised text have been marked in red.
Q: Line 60: "Colorectal cancer (CRC) is the second leading diagnosed cancer". Actually, CRC is the third most common diagnosis and second deadliest malignancy for both sexes combined [1].
A: This paragraph has been added in the Introduction section. (Page 2, Line 61)
Q: Line 61: "surgery and chemotherapy are used to treat CRC". Radiotherapy combined with chemotherapy is the standard of care in locally advanced rectal cancer in the setting of neoadjuvant treatment, while another modern aspect of CRC treatment is immunotherapy with PD-1 inhibitors, nivolumab and pembrolizumab, which currently constitute the new standard of care as treatment of chemotherapy-refractory MSI-high/MMR-d CRC [2].
A: This paragraph has been added in the Introduction section. (Page 2, Line 63)
Q: Line 89: "Phosphorylated" should be changed to "phosphorylation".
A: It has been corrected. Thanks reviewer’s assistance. (Page 3, Line 97)
Q: Line 164: Here the TAX and NOC are referred as microtubule-interfering agents (MIAs), whereas in the Introduction they are referred as microtubule-targeting agents (MTAs). While both terms are true and correct, one of the two should be applied in the whole manuscript for the sake of cohesion.
A: Both MIAs and MTAs have been used before. We applied microtubule-targeting agents (MTAs) to replace MIAs in the whole manuscript.
Q: Line 181-183: "In the same ... CRC cells". It should be mentioned that these effects are mediated by both TAX and NOC, although not inducing cdc2, as shown in Figure 2C.
A: The paragraph has been corrected as reviewer’s suggestions. Thanks. (Page 5, Line 189)
Q: Line 200: Replace "An" with "The".
A: It has been corrected. (Page 5, Line 208)
Q: Line 223: "was reduced". More accurate to state that it "was reversed", since the reduction is profound only in G2/M ratio.
A: It has been corrected. (Page 6, Line 231)
Q: Line 239: “suppressed by adding the PERK inhibitor GSK, or JNK inhibitor SP and JNKI (Fig. 7B, C)”. Although it is described here, the effect of JNKI is not depicted in Figure 7.
A: We have added JNK inhibitor (JNKI) in the legend of Fig. 7. (Page 17)
Q: Line 293: It is not JNK siRNA, but JNKI that reversed TAX- or NOC-induced apoptosis in CRC cells.
A: The mistake has been corrected. Thanks. (Page 7, Line 301)
Q: Line 451-452 and 464: “CRCs” should be replaced by “CRC cells”.
A: In the whole manuscript, “CRCs” has been corrected to “CRC cells”.
Q: Line 466 and 469: HT-29, not HCL-15 is depicted in Figure 2B,C.
A: Thanks reviewer’s suggestions. The mistake has been corrected. (Page 12, Line 517 and 520)
Q: Line 491: “increased in” should be replaced by “the increase in”.
A: It has been corrected to “the increase in”. (Page 14, Line 541)
Q: Line 508: HCT-115 should be replaced by HCT-15 in Figure 6
A: The mistake has been corrected. (Page 16; Fig. 6)
Q: Line 518: “cyc25C” should be replaced by “cdc25C”.
A: The mistake has been corrected. Thanks reviewer’s assistance.(Page 16, Line 572)
Q: Line 532-534: The legend of Figure 7B lacks of description of the corresponding results.
A: The description of Fig. 7B has been included in the legend. (Page 17, Line 587)
Q: In the Discussion section two small paragraphs should be added; one describing the limitations of the study, and one further expanding suggestions for future research.
A: A paragraph describing limitations of the study and suggestions for future research have been included in the Discussion (Page 8; Line 326)
Reviewer 2 Report
Nice work done by Dr. Chen and group elaborating the role of PERK in therapeutic aspect of cancer cells. the whole study is well designed and followed by the hypothesis. Few things should be addressed before it is ready for publication. they are as follows:
Few western blot signals are not that prominent, specifically for Fig 5. Authors should quantify the phospho proteins on basis of total proteins and indicate the values. They can indicate the quantified numbers next to the blots or represent in a graph and add it as a subfigure. For referral purposes can refer PMID: 25632961 for the quantification and normalization methods. While JNK is related to oncogenic KRAS signaling pathways PMID: 21282468, it has been shown that after glutamine starvation KRAS mutant cells also get arrested at S or G2/M phase of cell cycle and upon treatment of paclitaxel/ microtubule stabilizer PMID: 25023699. It will be interesting to see author's perspective on PERK's contribution in this theme. Authors should add few lines in the light of PERK/JNK/KRAS/cancer metabolism by referring the mentioned work as appropriated. It will help to broaden the significance of this study. It will be interesting see how JNK/PERK plays role in cell cycle check points. By referring PMID: 26682255 authors should also add few lines in discussion on this aspect. It will be provocative to design future studies on the basis of proposed layout. Authors opinion on this aspect will be beneficial for general audience in case therapeutic application.
Author Response
Response to Reviewer 2
Thank you for comments on our submission. Responses to reviewer’s comments are attached below, and all changes are marked in red in the revised text.
Q: Few western blot signals are not that prominent, specifically for Fig 5. Authors should quantify the phospho proteins on basis of total proteins and indicate the values.A: Quantitative analysis of Western blotting data in Fig. 5 has been done, and the relative levels of pPERK (Thr980) and pJNK were normalized to α-TUB and tJNK, data has been included in the revised manuscript. A reference (PMID: 25632961) has been cited here. (Page 15, Line 557)
Q: While JNK is related to oncogenic KRAS signaling pathways PMID: 21282468, it has been shown that after glutamine starvation KRAS mutant cells also get arrested at S or G2/M phase of cell cycle and upon treatment of paclitaxel/ microtubule stabilizer PMID: 25023699. It will be interesting to see author's perspective on PERK's contribution in this theme………….A: Additionally, Cellurale et al (2011) reported that JNK is required for Ras-induced transformation of MEF [48]. In cancer cells harboring K-Ras mutations, glutamine (Gln) deprivation induced cytotoxicity to TAX via arrested cell cycle at S or G2/M phase [49]. It indicated that K-Ras-driven cancer cells overrode a late G1 checkpoint by Gln deprivation, and arrested at S phase, exploiting the potential of metabolic changes in cancer cells for therapeutic intervention [50]. In the present, we found a cross-activation between JNK and PERK leading to apoptosis and G2/M arrest by TAX and NOC in human CRC cells. Contribution of PERK/JNK to K-Ras-mediated cancer metabolism and cell cycle progression is suggested for further study. This paragraph has been included in the Discussion section. (Page 7, Line 304)
Q: It will be provocative to design future studies on the basis of proposed layout. Authors opinion on this aspect will be beneficial for general audience in case therapeutic application.A: Our study gives a new aspect revealing PERK pathway as a possible target for CRC treatment, however the study was performed in a limited number of human CRC cells, and lack of in vivo study. Studying the actions of PERK in different CRC cells, and examining the expression of PERK in human CRC samples and CRC animal models related to their sensitivity to chemotherapy are important topics for further research. (Page 8, Line 327)
Round 2
Reviewer 2 Report
All concerns has been addressed and changes been incorporated in manuscript. Ready for acceptance.